

# On defining climate by means of an ensemble

Gábor Drótos[1,2,3] and Tamás Bódai[4,5,6]

[1]MTA-ELTE Theoretical Physics Research Group, Pázmány Péter sétány 1/A, H-1117 Budapest, Hungary
[2]Instituto de Física Interdisciplinar y Sistemas Complejos (CSIC-UIB), Campus UIB, Carretera de Valldemossa, km 7,5, E-07122 Palma de Mallorca, Spain
[3]HUN-REN Institute for Nuclear Research, Bem tér 18/C, H-4026 Debrcen, Hungary
[4]Pusan National University, Busandaehak-ro 63beon-gil 2 (Jangjeon-dong), Geumjeong-gu, 46241 Busan, Republic of Korea
[5]Center for Climate Physics, Institute for Basic Science, Busandaehak-ro 63beon-gil 2 (Jangjeon-dong), Geumjeong-gu, 46241 Busan, Republic of Korea
[6]Department of Applied Statistics, Institute for Mathematics and Basic Sciences, Hungarian University of Agriculture and Life Sciences, Budapest, Hungary

**Correspondence:** Gábor Drótos (drotos@general.elte.hu)

**Abstract.** We study the suitability of an initial condition ensemble to form the conceptual basis of defining climate. We point out that the most important criterion for this is the uniqueness of the probability measure on which the definition relies. We first naively propose, in harmony with earlier work, to represent such a probability measure by the distribution of ensemble members that has, loosely speaking, converged to the natural probability measure of the so-called snapshot or pullback attractor of the dynamics; this attractor is time dependent in the presence of external forcing. Then we refine the proposal by taking a probability measure that is conditional on the (possibly time-evolving) state of modes characterized by time scales of convergence that are longer than the horizon of a particular study. We discuss the applicability of such a definition in the Earth system and its realistic models, and conclude that the practically relevant probability measure may, hopefully, become accessible by a few decades of convergence after initialization; for this, initialization may perhaps need to rely on the observed state of the slower-converging modes. However, the absence of sufficient separation of time scales of convergence between modes or regime transitions in variables corresponding to slower-converging modes might preclude uniqueness, perhaps in certain subsystems. In uniqueness holds, time evolution of slower-converging modes may induce unforced climate changes, leading to the need for targeted investigations to determine the forced response. We propose an initialization scheme for studying all these issues in Earth system models.

## 1 Motivation and historical background

In the past decade or so, $\mathcal{O}(10)$–$\mathcal{O}(100)$-member initial-condition ensemble simulations in single state-of-the-art Earth system models have been run (e.g., Kay et al., 2015; Maher et al., 2019; Rodgers et al., 2021), and such simulations are expected to become widespread in the near future. The aim of these projects is the identification of forced responses and the exploration of the internal variability of climate under explicit time dependence (which the word 'forcing' will refer to throughout this article). The motivation for this kind of investigation is based on the naive recognition (e.g., Deser et al., 2020, and references therein) that the Earth system (and models thereof) permits a plethora of equally plausible states (e.g., weather configurations)



at any time instant of a forcing scenario[1]: the ensembles in question are intended to represent this set of states of the system, for which their different members have been generated with the same forcing scenario but stemming from different initial conditions. Climatological mean values are then identified with the ensemble mean, and it is more and more widespread to

describe the internal variability of climate by further statistical quantifiers evaluated with respect to the ensemble; in turn, the time evolution of all these statistical quantifiers (including but not restricted to the mean) is usually identified with their forced response.

One assumption behind these interpretations is that climate is defined through the distribution of the ensemble members. However, in case this distribution is not unique but depends on personal choices (e.g., on the precise time and way of initializa-

tion), every research group creating an ensemble simulation may happen to define its own climate even using the same model and forcing scenario within the same predictable context (which we define and explain later): such a climate is not characteristic to the system, the forcing scenario and perhaps other objective factors, defining the predictable context, alone. Due to the dependence on the aforementioned personal choices, it is a "personal climate", subjective in its nature, which should be avoided (cf. Werndl, 2019). An aim of this article is a deeper investigation of this issue.[2]

Traditionally, specific definitions of climate, beyond identifying it with a statistical description of the "climate system" or the Earth system, were usually formulated in terms of *temporal* statistics of weather (e.g., [author field is not applicable], 2021), a practice that still determines the thinking of many researchers from various relevant fields (Flandoli et al., 2022; Nicklas et al., 2022). If no forcing acted on the Earth system, i.e., if its equations of motion (including boundary conditions) did not depend explicitly on time, the interval for evaluating temporal statistics could be extended to the infinite future, and these

statistics would coincide with those evaluated with respect to an infinite-size ensemble distributed according to the natural probability measure of the chaotic attractor of the system (Ott, 1993) [i.e., Birkhoff's ergodic theorem would hold; note that the Earth system is dissipative, and can generally be assumed to be chaotic (Herein et al., 2023)]. The same would be true for a temporally periodic forcing handled in terms of a stroboscopic map (Tél and Gruiz, 2006). However, if the forcing is not periodic, such a construction is not available. In that case, statistical quantifiers evaluated with respect to time need not

correspond to any probability measure that would be relevant to any particular time instant (Drótos et al., 2015) [this is most easily seen if the system is forced by a monotonic parameter drift, cf. Jánosi and Tél (2024)].

While Lorenz recognized the phenomenon (namely chaos) that renders the evolution of the state of the system unpredictable (Lorenz, 1963), and also formulated that the permitted states of the system may be represented by an ensemble of realizations that differ in their initial conditions (Lorenz, 1975), his notion of climate still assumed that the statistical descriptors are

approximately constant for shorter or longer times (Lorenz, 1975). At the same time, Hasselmann (1976) was thinking about a statistical mechanical analogy that implied the relevance of an ensemble but without justification. Leith (1975a, b, 1978) worked out the same statistical mechanical analogy in more detail and also including the change of the ensemble statistics as a response to some external forcing (but assuming slow variation of the latter). He was explicitly motivated by the unpredictability of the individual realizations, but he did not discuss how the distribution of the ensemble members is determined and if it is unique.

---

[1]We shall use the term 'scenario' to refer to the form of time dependence of the forcing, irrespective of whether past or future forcing is considered.

[2]Note that the forced response of climate, which will also be discussed, can only be defined if climate itself is defined.




Branstator and Teng (2010) revived the latter picture, allowed for variations in the forcing on any time scale, and also assumed (but not justified) the uniqueness of a distribution, already called *climatological* distribution, emerging from arbitrary initial conditions in the infinite past. This climatological distribution gave the reference basis for ensembles of trajectories (or abstract probability densities) initialized later, by which predictability was studied. Finally, DelSole and Tippett (2018) adopted their approach, discussed uniqueness in dynamical systems, and demonstrated it in a stochastic model, providing

thereby a well-established notion of the "standard" climatological distribution. Also interested in predictability, however, they were not satisfied with this notion, and introduced a precise probabilistic framework that based a climatological distribution on observations of the more recent past. By such a definition, the unique nature of the concept of climate is again lost; see Appendix A for further details.[3]

The same issue concerns the approach of Stainforth and coworkers (Stainforth et al., 2007; Daron and Stainforth, 2013;

de Melo Viríssimo et al., 2024). They identified a changing climate early on with the image of an attractor traced out by an ensemble (Stainforth et al., 2007) and also emphasized the uniqueness of a distribution related to the attractor under stationary conditions (Stainforth et al., 2007; Daron and Stainforth, 2013). On the other hand, they aptly recognized that narrowing down initial uncertainty (the breadth of an initial distribution) in variables characterized by (some kind of) time scales that are long compared with the time span targeted by a particular study will result in a more relevant distribution than the full distribution

permitted by the system (Stainforth et al., 2007; Daron and Stainforth, 2013; Hawkins et al., 2016); a similar idea shall be extensively discussed in Section 3. However, they finally took "future climate as a distribution conditioned on our uncertain knowledge of the system's current state (Stainforth et al 2007)" (Daron and Stainforth, 2013) [a view implicitly adopted in Hawkins et al. (2016) as well], similarly as DelSole and Tippett (2018), losing thus uniqueness. Their most recent work (de Melo Viríssimo et al., 2024) still aligns with this view but in the explicit context of the snapshot/pullback framework (to be

discussed below). They included here a major critical remark about this framework, which, however, had already been pointed out by a reviewer of Drótos et al. (2015) and which we had already addressed in an earlier preprint version of the present article (Drotos and Bodai, 2022).

Actually, Ghil et al. (2008); Chekroun et al. (2011) already drew attention to a climatological relevance of the concept of pullback attractors and the natural probability measures supported by them, corresponding to a unique probability density to

which other densities converge at a given time $t$ if their initialization time $t_0$ tends to $-\infty$ in a dissipative chaotic system even with temporally varying parameters. This unique density was shown to be traced out by an ensemble of trajectories initialized in the remote past. However, Ghil et al. (2008); Chekroun et al. (2011) and related work (e.g., Ghil, 2014; Pierini et al., 2016) applied this concept to subsystems of the Earth, which were subject to randomly generated forcing with a fixed distribution instead of a drifting signal. Even the review by Ghil and Lucarini (2020) hardly mentions the latter possibility. Although

Werndl (2016) applied the concept of pullback attractors to general forms of forcing, including drifting ones, she discarded the corresponding definition of climate. Flandoli et al. (2022) did not restrict the form of forcing either, but they based their analysis

---

[3]Note that loss of predictability and convergence to a unique distribution are two facets of the same phenomenon: climate should, basically, be identified when predictability is lost.





on a slow time dependence compared with the convergence of time averages to ensemble averages, which is generically not the case and is certainly not so during the ongoing global warming.

In fact, it was in Bódai and Tél (2012) that the applicability of the approach of Ghil et al. (2008) to the description of global climate changes was first recognized by writing that "climate change can be seen as the evolution of snapshot attractors" (note also that the correspondence between the rigorously defined pullback attractor and the so-called snapshot attractor, which was introduced to the physics literature by Romeiras et al. (1990), was pointed out there). We discussed in detail in Drótos et al. (2015) that the uniqueness of the natural probability measure of a pullback or snapshot attractor (as represented by an initial-condition ensemble, although in a toy model) makes it the appropriate concept for describing the statistics, i.e., the climate, of a system forced by a drifting signal (both in terms of the "mean state" and in that of the internal variability, both of which respond to a forcing).

One of the most important numerical observations of Drótos et al. (2015), already formulated there in the text but in a preliminary form, is about a convergence towards an actual (rigorously defined) snapshot/pullback attractor during *forward time evolution*, i.e., in a push-forward sense [as opposed to the pullback sense which underlies both a pullback (Ghil et al., 2008) and a snapshot (Namenson et al., 1996) attractor and means tending to the remote past, eventually to $-\infty$, with the time of *initialization* $t_0$ while keeping the time instant of interest, $t$, fixed]. The particular observation is that forward convergence progresses in an approximately exponential way after some kind of transients, so that the actual snapshot/pullback attractor is arbitrarily approached within some "short" time. Therefore, constructing an ensemble numerically at the time instant of interest, $t$, such that it represents the snapshot/pullback attractor and its natural probability measure at $t$ with a "sufficient" accuracy does *not* require initializing the ensemble in a very much remote past. Instead, initializing a few approximate $e$-folding times earlier is usually satisfactory, and convergence is ensured by (and can be monitored during) forward time evolution.[4]

In Herein et al. (2016) the applicability of this framework to an intermediate-complexity general circulation model was illustrated, which was also used in Herein et al. (2017) to extend the framework to variables describing spatial patterns. In Drótos et al. (2017) we also pointed out the practical relevance of the approach in the same model by investigating the convergence to the density of the natural probability measure from initial conditions obtained by a slight perturbation of a state of the system already located on the attractor. The conceptual power of the snapshot/pullback framework was underlined by Vincze et al. (2017) by applying it to a laboratory experiment. An overview of basics and applications of this framework to describe climate is provided in Tél et al. (2020), where previously disregarded issues are also discussed.

In this article, it is first explained why the description based on the natural probability measure of an attractor could actually be regarded as a natural definition of climate, then a caveat of this definition is reiterated from above, namely that unpredictable long-time-scale internal variability of the full system is normally desirable to be excluded [cf. the Edinburgh effect in Stainforth (2023) and Box 1 in Lucarini and Chekroun (2023)]. To overcome this caveat, we propose an improved definition, which relies on a hypothetical probability measure that is conditioned on the state of modes identified to be slow in a certain sense. Whether this improved definition is directly applicable to the real Earth system and its fully coupled models will depend on the degree of separation between time scales of convergence; some hope for an affirmative answer is provided by existing analyses (e.g.,

---

[4]As a further implication, a forward-time limit of $\infty$ is not relevant; in fact, the forcing or the system need not even be defined in this limit.





Li and Jarvis, 2009; Olivié et al., 2012) suggesting a possible gap between convergence taking place in a few decades and a few centuries. In case the separation proves to be too small, our proposed definition may still serve as a guidance for how to theoretically treat probabilistic aspects and how to design practically useful constructions. A closer look at the concept of forced response will also be taken, highlighting that a climate conditioned on the state of slow modes can change without the
introduction of a forcing.

Note that the process relevant for defining climate is described by terms of a sum with each term converging in time in an exponential-like manner, so that any general conclusions of this study, which do not consider particular models and particular variables, must remain qualitative and leave room for more concrete findings resulting from follow-up work.

## 2 A naive proposal for the definition of climate

In this section, we will illustrate the basic idea through an intermediate-complexity climate model, without drawing final conclusions about the real Earth system or realistic models thereof. The particular intermediate-complexity model is the Planet Simulator (PlaSim) (Fraedrich et al., 2005) with a mixed-layer ocean. We use the same model output as in Drótos et al. (2017); please refer to this publication for more details about the configuration.

For this section, let us imagine that the dynamics of the Earth system is described perfectly by PlaSim in the mentioned
configuration (including even the discretized nature of the model). This defines our dynamical system, which has a phase space of $\approx 10^5$ variables. Let our model Earth system be subject to the following forcing scenario:

$$[\mathrm{CO_2}](t) = \begin{cases} 360 & \text{for } t < 600, \\ 360 + 3.6(t - 600) & \text{for } t \geq 600, \end{cases} \tag{1}$$

where concentration is in ppm and time is in yr, and let us study the time evolution of the annual mean near-surface temperature at a particular grid point in the southern Pacific Ocean [similarly as in Drótos et al. (2017)]. Finally, let us suppose that our
Earth system has followed the trajectory corresponding to the dark gray line of Fig. 1 up to $t_0 = 610$yr, which we identify with the present.

A possible question about the *weather* of the future is, to take an (almost) arbitrarily chosen example, say, what the near-surface temperature at the given grid point will be at $t_1 = 694$yr. To obtain this temperature, we can use the $t = t_0$ phase space position of the dark gray trajectory of Fig. 1 as initial condition, and integrate PlaSim from $t_0$ to $t_1$: the result for the
near-surface temperature of the given grid point is the red trajectory shown in Fig. 1, which has, of course, a unique value at $t_1$. However, this value is only one possible answer to the question.

Let us slightly perturb the $t = t_0$ phase space position of the dark gray trajectory 192 times to obtain 192 different initial conditions: the utilized random perturbation modifies the surface pressure field on the order of $10^{-3}$hPa [see again Drótos et al. (2017) for the details], so that each perturbed initial condition can be regarded as realistic as the original one. In particular, these
different initial conditions could not be distinguished by standard instrumental measurements on the real Earth. The integration of PlaSim from these initial conditions results in an ensemble of trajectories, plotted in light blue in Fig. 1 (only 48 of the 192

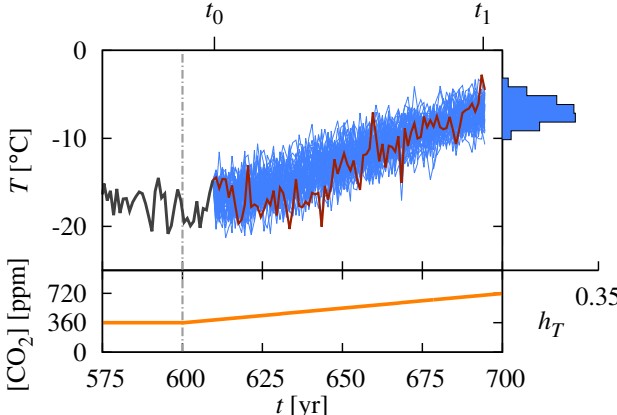

**Figure 1.** The annual mean near-surface temperature $T$ of a single grid point in the southern Pacific Ocean (at $180°$E and about $64°$S) as a function of time. $t_0 = 610$yr is identified with the present, and $t_1 = 694$yr is an (almost) arbitrarily chosen year in the future. The dark gray line is a single PlaSim simulation and is regarded as the instrumental record in the model system. The red line is the continuation of the same simulation and is regarded as a prediction. The 48 light blue lines are alternative predictions, obtained from simulations initialized by slightly perturbing the surface pressure field of the simulation of the dark gray line at $t_0 = 610$yr [see the main text and Drótos et al. (2017) for details]. The normalized histogram $h_T$ constructed from the 48 values of the light blue lines and further 144 alternative predictions (a total of 192 values) at $t_1$ is shown on the right-hand-side of the main plot. The $CO_2$ concentration, through which the forcing scenario is defined, is also displayed (in orange). The vertical dot-dashed line in gray marks the beginning of the linear ramp in the $CO_2$ concentration. Data are from Drotos (2022).

for better visibility), each giving a separate answer for the *weather* at $t_1$. All are possible answers, and the differences between them emerge as a result of the chaotic nature (unpredictable internal variability; Herein et al., 2023) of the system.

In fact, infinitely many of these answers would trace out a density, as indicated by the normalized histogram $h_T$ in Fig. 1. We
can ask now if this density provides a relevant characterization of the plethora of all possible near-surface temperature values at $t_1$.

In particular, the question is about the uniqueness of this density. If we perturb the original initial condition at $t_0$ in a different way, will we end up with a different density at $t_1$? For example, if we take a larger or a smaller magnitude for the perturbation, will the final density be broader or more narrow? Or if we restrict the sign in the perturbation, will it shift the final density in
some direction? Since the initial conditions can never be constrained to arbitrary precision, such perturbations can be regarded just as relevant as the original one.

In fact, for an infinitely large ensemble in a dissipative nonautonomous dynamical system exhibiting chaotic behavior (such as PlaSim), the kind of density in question corresponds to the natural probability measure of a snapshot (Romeiras et al., 1990) or pullback (Ghil et al., 2008; Chekroun et al., 2011) attractor if $t_0$ tends to $-\infty$, at least as long as the initial conditions remain
in the basin of attraction of the same attractor[5]. The density of the natural probability measure is unique, i.e., it will be the same

---

[5]It is assumed here that the relevant basin of attraction existed in the infinitely remote past; cf. Section 3 and Footnote 14 in particular.



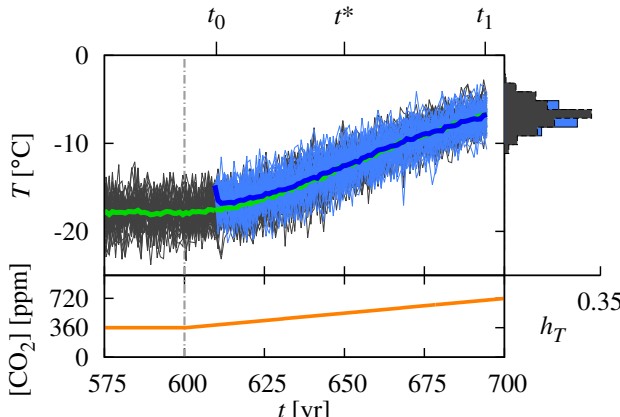

**Figure 2.** The light blue lines and histogram of Fig. 1 compared with an ensemble of the same size (marked by dark gray) initialized in the remote past (at $t = 0$, asymptotically far in practice from $t = 575$). The corresponding ensemble averages are also shown in dark blue and green, respectively. By $t^*$, the two ensemble averages practically coincide. Data are from Drotos (2022).

for *any* generic set of infinitely numerous initial conditions within the basin of attraction of the given attractor. As we discussed in the introduction, an atmospherically motivated toy model was used in Drótos et al. (2015) to illustrate that a process of convergence to the natural probability measure (*forward in time*) is exponential-like on the long term (it is presumably faster than any power law, cf. Appendix B). In such a case, the convergence of a finite-size ensemble is practically accomplished

(with "exponential precision", according to some practical point of view; see later) within a finite amount of time. In Herein et al. (2016), this was found to be the case for PlaSim, too, and this finding was conjectured to be relevant to any global climate model or Earth system model. In our configuration of PlaSim, the convergence time proved to be a few decades.

     As a consequence, the *probability* density of the near-surface temperature at the investigated grid point in our configuration (and that of any other variable or set of variables) that is traced out by even a finite-size ensemble at $t_1$ will be practically the

same regardless of how we choose initial conditions at $t_0$ — provided that two conditions are met. The first one is that $t_0$ has to be sufficiently far, although not infinitely far, in the past from $t_1$. The second is that the initial conditions must be chosen within such limits that ensure convergence to the desired attractor, avoiding a different one, e.g., a snowball Earth (cf. Kaszás et al., 2019; Ragon et al., 2022). Since $t_0$ and $t_1$ are separated by several decades in Fig. 1, this conclusion translates for our example as follows: within the mentioned limits, we cannot modify the initialization scheme at $t_0$ to obtain a substantially

different probability density at $t_1$; that is, within a certain range of initialization schemes, we cannot "ruin" the quantitative result. Among other initialization schemes (which need not rely on some particular trajectory of the system), an extremely small perturbation of the actually realized initial conditions (the $t = t_0$ phase space position corresponding to the dark gray line of Fig. 1 in our model Earth system, the equivalent of hypothetical perfect instrumental observations at $t_0$ in the real Earth system) will lead to this unique density, which corresponds to the natural probability measure, see Fig. 2.





These considerations suggest this unique density (the 'natural probability density' in what follows) to be the practically relevant *a priori* (Paillard, 2008) probability density (i.e., existing independently of almost *any* observation about the system) that can be associated with the plethora of all possibilities at a future time instant $t_1$ that are permitted by the dynamics under the given forcing scenario (within the relevant basin of attraction, being the only observational constraint). That is, beyond being of practical interest, the natural probability density basically characterizes the *system* in statistical or probabilistic terms,

instead of characterizing a particular situation (a "microstate" in a statistical mechanical analogy, e.g., a weather configuration in the atmosphere[6]) permitted by the system. Note that this abstract density exists at *any* time instant: not only at $t_1$, but also after and even before it (for instance, also at the time instant $t_0$ identified with the present, or even earlier; see the dark gray ensemble of Fig. 2). To practically obtain this density at a given time instant, one just needs to prescribe initial conditions for an ensemble in the sufficiently far past (within the relevant basin of attraction), and follow the time evolution of the ensemble

until the desired time instant.

Of course, the natural probability density depends on time in the presence of a forcing (which is the generic situation and is so in the dynamics of the real Earth system; see again the dark gray ensemble of Fig. 2 for our model configuration). Otherwise, it is constant, and coincides with the natural probability density of the usual chaotic (stationary) attractor of the dynamical system (Ott, 1993).[78]

We now recall that any definition of climate intends to capture statistical properties (including those related to temporal aspects). We also see that the relevant statistical properties of the system are, at root, described by the natural probability measure of the relevant snapshot or pullback attractor at any time instant. Therefore, *we would hereby naively suggest defining climate as the statistical properties determined by the (infinitely large) ensemble of trajectories evolving according to the natural probability measure under a given forcing scenario.* Then the expected value of some given variable will be the climatic mean value of that variable, and all higher-order moments will describe internal variability. With this definition, any particular realization

of the dynamics performs a sampling of the probability density that defines climate [i.e., the 'climatological distribution' in the terminology of Branstator and Teng (2010)].

---

[6]The prefix 'micro' is not to be confused with that used in the term 'microinitialization' where it refers to a certain type of (in particular, fast) variable which is perturbed (Stainforth et al., 2007).

[7]Note that the natural probability density should actually be defined in the full phase space of the system. The probability density of a given variable (such as the near-surface temperature at the selected grid point in the example of Figs. 1-2) is the marginal density of this multivariate probability density. The full multivariate probability density carries information about the statistical relationships between different parts of the system (different variables, different geographical regions, etc.).

[8]As a subtlety, the natural probability measure is defined instantaneously in mathematical terms, whereas our numerical example considered an annual mean. This apparent discrepancy is easy to resolve. On the one hand, the individual time evolution of the trajectories composing the ensemble results in their phase space positions to be distributed according to the natural probability density at any time instant; on the other hand, since a particular solution of a dynamical system is unique, the time evolution of the trajectories also uniquely defines a probability density for temporal averages evaluated along these individual trajectories. (In fact, this is true not only for averages, but for any quantity derived from an interval of time or simply from more than one discrete time instant.) The latter construction was termed "interval-wise taken" in Drótos et al. (2015). For the annual mean near-surface temperature of our example, this density is what we called the natural probability density. We thus see that generalization to some finite time interval of interest (days, months, years, etc.) is straightforward.



## 3 Applicability in realistic modeling; a conditional definition

We will now explore the extent to which the above ideas can be applied in realistic modeling studies with practical aims. We
will find an important implication but also that a refinement of the definition is necessary in the presence of long time scales of
convergence compared with the time span of a particular investigation, and point out further possible issues.

In practical numerical modeling, the natural probability density is approximately represented by the distribution of a finite-
size ensemble. Although we based our considerations for constructing the definition on the numerical example of our PlaSim
configuration, any global climate model or Earth system model shares the most relevant properties: they are nonautonomous
dissipative dynamical systems. Due to discretization, they have a finite number of variables, but, on the one hand, it is plausible
to think that a system described by partial differential equations should be obtained as a limit of infinitely many ordinary
differential equations and thus exhibits similar behavior; and, on the other hand, spatial autocorrelation reduces the effective
number of degrees of freedom to a finite value. As a consequence, our considerations can presumably be extended both to other
models and to the real Earth system. Cf. Lucarini and Chekroun (2023).

From the point of view of numerical modeling, however, the nonzero difference between the light blue ensemble and the
natural probability density (represented by the dark gray ensemble) approximately up to $t^* = 650$ in Fig. 2 implies a warning:
before convergence to the natural probability density takes place up to an accuracy prescribed with regard to some practical
perspective, a numerical ensemble of trajectories does *not* represent the statistical properties relevant for climate according to
the criteria discussed above (Drótos et al., 2017). Climate-related studies should, in principle, utilize ensembles for which such
a convergence has taken place. [With a term frequently used by the modeling community, the 'ensemble spread' must already
correctly describe the natural probability density with an accuracy imposed by some practical aspect; cf. Deser et al. (2025).]

Unfortunately, the practical situation may be unfavorable from this point of view: the issue of nonzero convergence time
poses a conceptual problem if long time scales of convergence exist in comparison with the horizon of a particular study. As a
prominent example in the real Earth system and in fully coupled models thereof, time scales up to the order of 1000 years may
appear in association with the deep ocean (Li and Jarvis, 2009), while studies with immediate practical relevance concentrate
on the last and forthcoming few centuries.[9]

In such a situation, one would presumably not wish to take the full (unpredictable) variability introduced by these slow
processes into account (see the Edinburgh effect in Stainforth, 2023). In the ideal case of an infinitely large separation between
the long and the targeted time scales (which implies that there must be an infinitely large gap in the spectrum of characteristic
convergence time scales of the system), one might use a single realization (in the simplest case: a single value, a single
"microstate") in variables associated with the long time scales ('slow variables' in what follows). This single realization ideally
should, of course, be based on instrumental observations in most studies [cf. Stainforth et al. (2007); Daron and Stainforth
(2013); DelSole and Tippett (2018); de Melo Viríssimo et al. (2024) and Appendix A]. Using a single realization would lead to
a *conditional* probability measure for the rest of the system, and a corresponding *conditional definition* of climate (cf. Hawkins
et al., 2016). In the simplest case when there is no fast but predictable evolution in such variables either[10], *this would be the*

---

[9]For some theoretical background for identifying different time scales of convergence, see Appendix B.

[10]See again Appendix B for some background about this issue.





*natural probability measure of the subsystem obtained by fixing the values of the slow variables* and would thus be unique (within a given basin of attraction). Then, initialization of an ensemble by one single or several but well-localized values of the slow variables (usually representing some instrumental observations[11]) will be the appropriate approach as well if the separation between fast and slow time scales of convergence is not infinitely but "sufficiently" (see later) large; still, these

values will remain unchanged in the simplest case.

Whether we are facing this simplest case or not depends on time scales that are, however, generally different from those of convergence. In general, "slow variables" may very well evolve fast (in terms of their time derivatives or how quickly they explore their range of possible values), the question is whether or, more precisely, for how long their evolution remains *predictable*. [This is reflected, for instance, in their temporal autocorrelation function, with the warning that it has two time

arguments in a nonautonomous case (Tél et al., 2020); see also Herein et al. (2023) for a discussion of the rate of growth of uncertainty, which is, at the same time, described in our context by the time scales of convergence as per Appendix B.] In case slow variables (defined in terms of time scales of convergence) do exhibit (practically) predictable evolution within the time span of a particular study, their values must not be kept constant, of course; instead, allowing for their time evolution is necessary when theoretically defining or numerically simulating climate, after an appropriate initialization (according to the

previous paragraph). A sufficient separation of time scales of convergence still ensures fast convergence to a unique probability measure but which is *conditional* on the state (initialization and subsequent evolution, which we suggest to call the predictable context) of the slow variables and thus itself evolves in time (even in the absence of forcing, i.e., explicit time dependence in the equations of motion). Note that explicitly identifying slow variables is not at all necessary; in case initialization is performed by applying a small perturbation to a (model) state (phase space position) involving any arbitrary variables, we will end up with

the desired probability measure after the mentioned fast convergence. (At wish, one can perhaps call this measure a 'conditional natural measure'.) The only point still requiring attention is which (model) state we choose for initialization.

Of course, 'unique' and 'sufficient' should be defined through some practical criterion for accuracy[12], as even a fast convergence never becomes fully complete but only in an asymptotic limit (assuming that such a limit is meaningful, cf. Footnote 4). Such a criterion may perhaps need to be determined by particular applications, but the corresponding definition of climate is

expected to be rather robust against this because of the exponential-like nature of convergence (still as per Appendix B).

We must acknowledge that the separation may perhaps happen not to be sufficiently large. Then it might not be possible to define a conditional probability measure that would be unique, neither mathematically nor in practice. DelSole and Tippett (2018) recognize the problem of possibly unseparated time scales and suggest to resolve it by conditioning the definition of climate on *observations* of the past; see Appendix A why this may not be fully satisfactory.

A further problem is posed by regime behavior (e.g., Franzke et al., 2015) in the slow variables even if the separation of time scales of convergence is sufficiently large from a practical point of view for most initializations. In particular, if initialization takes place during or shortly before a regime transition (such that convergence does not become "complete" before the regime

---

[11]Achieving an appropriate representation is technically non-trivial, cf. the phenomenon of initialization shock (e.g., Doblas-Reyes et al., 2011).

[12]The required accuracy can be defined through some threshold in some quantifier of distance; e.g., absolute difference in ensemble means and/or standard deviations, a Wasserstein distance (Panaretos and Zemel, 2019), or the Kullback–Leibler divergence (in fact not a metric) or some related quantifier (van Erven and Harremos, 2014).





transition), the evolving probability density may become strongly dependent on its initial condition even on a short time scale (in the sense of how it is distributed between the two regimes, i.e., how many ensemble members fall into one regime or 275 the other in a numerical investigation); and, according to the slow convergence, uniqueness may only be reached on the time scale of the slow variables. Thereby, uniqueness and thus a sound definition of climate will be lost on a short time scale. See Appendix C for a numerical illustration in a two-variable toy model where internal variability is modeled by stochastic terms.[13][14]

If slow processes (those that are associated with a convergence slower in comparison with the time span of interest as 280 identified according to Appendix B; slower-converging modes in what follows) have an influence on the time evolution of the practically relevant probability density, i.e., one that has already converged, we must emphasize that this is, if uniqueness is preserved, a climate change (according to the conditional definition of climate) but is not (entirely) a forced response [as it does not originate (only) from an explicit time dependence of the equations of motion]. That is, the two concepts delineate in such situations. A forced response can only be identified relative to the time evolution observed in the absence of any time 285 dependence in the relevant terms of the equations of motion. This is analogous to the practice of trying to remove spurious model drift (Gupta et al., 2013) in order to accurately determine trends of forced change. The implicit notion of an unforced climate change induced by variations in slower system components already exists, see, e.g., the relatively early work by Hawkins et al. (2016), especially points 2 and 4 in their summary section; cf. the discussion about possible gaps in the spectrum of the relevant operator in our next section. Note that Hawkins et al. (2016) also contains hints to possible answers to our 290 questions about predictability.

## 4   Time scales and system components; further practical issues

The main focus of our discussion here will concern the question of how the above-mentioned issues manifest themselves and how they may be related to individual system components in the real Earth system and its realistic models.

First, since a major separation of time scales of convergence is not obvious in the real Earth system and in realistic models 295 thereof, at present we are not able to assess how precisely the (possibly conditional) definition based on the natural probability measure of a snapshot attractor is applicable to these systems. The concept nevertheless gives guidance to decide what can be regarded climate and what cannot in a given study or some wider field of research.

---

[13]An even further issue can be the presence of multiple stationary chaotic attractors with intertwined basins of attraction, possibly including riddled basins (Alexander et al., 1992), and rate-dependent tipping (Ashwin et al., 2012) between them. In such a case, different initial densities will not converge even if their difference is relatively small, i.e., uniqueness will again be lost.

[14]One should also note that a chaotic snapshot attractor can split in the case of an underlying rate-dependent tipping (Kaszás et al., 2019), and only one of the branches will be relevant if observations for initialization are available after the splitting. Although the individual branches do not have their own, separate basins of attraction in the infinitely remote past, one can select those ensemble members that end up on the relevant branch, as in Kaszás et al. (2019). Even if the splitting takes place in the future, it is more natural to regard the two branches as two separate climates. The probability of ending up on a given branch is determined by the natural measure (or its conditional variant) at the time of the splitting, provided that initialization is performed sufficiently far in the past before the splitting. Otherwise, the determination of the probabilities is not unique.





This is illustrated by the example of the ocean, which is one influential source of long time scales in the Earth system. Studies about *response time scales* (an even further but related property) suggest that time scales associated with the mixed layer and with layers below the thermocline are separated by a factor of about 10, and there might be no other characteristic time scale in between (e.g., Li and Jarvis, 2009; Olivié et al., 2012).[15] Importantly, dynamical modes such as the Atlantic Meridional Overturning Circulation (AMOC; Buckley and Marshall, 2016) do not seem to introduce complications, see later. Such a separation might or might not be just enough to treat the deep ocean separately from the rest of the system: by the time "complete" variability would unfold in the mixed layer (in terms of its own time scales), possibly unpredictable variations in the deep ocean might just become large enough to be relevant for an appropriate description.

For an appropriate assessment of the issue, we need to reiterate that reaching "complete" variability must always be defined in terms of some practical criterion, according to the exponential-like nature of convergence. The distance from a suitably defined probability density may generally be thought to become negligible after a few times the longest relevant time scale (approximate $e$-folding time) passes from initialization.

In the particular case of the ocean, as mentioned, one characteristic time scale of the unfolding of variability can be supposed to be a few decades according to the results from Li and Jarvis (2009); Olivié et al. (2012), which has recently been explicitly confirmed by Deser et al. (2025) with regard to AMOC: the authors report a *convergence* time (until convergence becomes practically complete) up to about 40 years. Therefore, even conditional definitions of climate must rely on this unfolding in century-long studies of the Earth system [as pointed out by Deser et al. (2025) as well]. In terms of the definition of DelSole and Tippett (2018), a lead time of a few decades is required at least, and any shorter choice will inhibit a satisfactory interpretation of corresponding results. Whether the unfolding of (unpredictable) variability on longer time scales is required to be taken into account remains an open question. A separation by a factor of 10 gives quite some hope that the answer is 'no', although it is unclear why the intermediate time scale of the Atlantic Multidecadal Oscillation/Atlantic Multidecadal Variability Delworth et al. (e.g., 2007), if such a mode exists (Vincze and Jánosi, 2011; Mann et al., 2021), does not appear in the referenced analyses and what role it plays in the unfolding of variability. We emphasize it here that the numerically observable time scales may depend on the choice of the variable.

Actually, one may also think of situations with little correlation between fluctuations of different system components, for which an example might be the relationship between the deep ocean and the surface-related processes, in which case the unfolding of deep oceanic variability would be irrelevant for most observables of practical interest. According to recent research (Singh et al., 2022), there are some signs that suggest this to be so for most of the globe but not for the Southern Ocean.

To be precise, time scales of convergence may not necessarily be associated with given system components or variables considered explicitly in the equations of motion. In this respect as well, one should rely on the framework discussed in Appendix B, based on the spectral theory of transfer operators (Györgyi and Szépfalusy, 1988; Lasota and Mackey, 1994; Froyland et al., 2010; Chekroun et al., 2014; Slegers, 2019; Chekroun et al., 2020; Navarra et al., 2021). This concerns also the degree of

---

[15]Note that an apparent continuous dependence of the time scale on depth (Yang and Zhu, 2011) may well be a spurious result of an interplay between the two mentioned time scales. Furthermore, the smaller separation identified by Hogan and Sriver (2019) may originate from a suboptimal partitioning of the water column from the presently discussed point of view.





correlation between the internal variability of different system components; although, as explained in the previous Section, this
is presumably relevant only to how a (model) state should be chosen for initialization (attending to the predictable context).

One also has to consider the possibility that the real Earth system or some realistic model thereof does *not* meet the pre-
requisites of the description of convergence described in Appendix B. Long-term persistence (polynomially decaying auto-
correlation) and, more generally, scaling of fluctuations in climate-related time series are reviewed by Franzke et al. (2020).

As pointed out in their sections 2.3 and 5, long-term persistence and scaling may be illusory or they may result from external
forcing, which would be in harmony with the Markovian nature of (most of the) equations of motion.[16] In any case, if there is a
break in the power spectrum such that there is a non-scaling regime of considerable length beyond the break (as, e.g., in Vincze
and Jánosi, 2011), the decomposition of the convergence to (a generalization of) eigenmodes (as detailed in Appendix B) may
retain its pertinence.

Before summarizing, we need to take care about the issue of regime behavior as raised in Section 3. Actually, the absence
of different qualitative behaviors in global variables between members of currently existing large ensembles suggests that such
an effect is restricted to particular system components at most. Such an effect might nevertheless appear in some slow system
components [possible examples are related to Labrador Sea ice cover (Danabasoglu et al., 2020) and Southern Ocean variability
(Gnanadesikan et al., 2020)], and uniqueness might or might not be lost in these cases: it depends on the way or the time of

initialization. Cf. the illustration in Appendix C.[17]

Taken together, we can say that distinguishing between different time scales of convergence appears to hopefully provide
a sound definition of climate in the sense of Section 3: *from the point of view of century-long investigations, we might hope
to be able to meaningfully define climate through the probability measure obtained after a convergence time of a few decades*
[perhaps up to four, cf. Deser et al. (2025)]. Further gaps in the spectrum of convergence time scales may possibly give rise to

sound definitions for investigations on other time scales as well, but whether such gaps exist is an open question at present.

While the potential relevance of separation between time scales in formulating a practically satisfying definition of climate
was already discussed by Lorenz (1975) and was maybe born together with the very first attempts to define climate, please
note that time scales of *convergence* are concerned in the conditional definition formulated in Section 3. As a consequence,
this definition allows external forcing to induce climate changes on *arbitrarily short* time scales (e.g., after volcanic eruptions),

irrespective of possible slower modes of internal variability.

---

[16]Note that the so-called Hurst effect does not imply long-term persistence (Franzke et al., 2015).

[17]Intertwined basins of attraction could have an effect very similar to that of regime transitions. However, we regard them unlikely to play a role if
initialization is performed by means of a small perturbation of a state, i.e., phase space position, on an attractor.



## 5 Conclusions; proposal for an initialization scheme

### 5.1 What we can conclude about

As a main conclusion, an operational definition of climate, in the context of centennial studies at least, might rely on a decadal-scale convergence of an ensemble[18] within a basin of attraction to a (practically) unique but time-dependent probability density: this density could be identified with climate. In case the state of slower-converging modes have a considerable relevance in determining this probability density, which thus becomes conditional on that state (i.e., the predictable context), this definition assumes

**(a)** a sufficiently large separation between the relevant time scales of *convergence*, and

**(b)** the avoidance of a regime transition in association with the given slower-converging modes at initialization.

For such a case, we spell out two issues that require further attention.

**(i)** For climate projections, initialization should rely on the observed state of the slower-converging modes (which is most easily achieved by using observations to initialize the whole system). Note that this is not so in current practice, but initial conditions in terms of slower-converging modes rely on one or more arbitrary time instants of a long control run: this may be problematic for the purpose of preparing climate projections regardless of how climate is defined. On the other hand, such a sampling is important from a more general point of view.

**(ii)** The concept of climate change may delineate from that of a forced response, i.e., climate change may become partially unforced.

We emphasize that the above definition meets our criterion about uniqueness described in Section 1: it only depends on the system, the forcing, and the predictable context. The latter is provided (in fact, together with the basin of attraction) by the state of the slower-converging modes and is objective from the point of view of century-long studies since, by definition, slower-converging modes remain sufficiently *predictable* within this time span so that one can learn their state in principle. Should condition (a) be violated, observations of slower-converging modes will not provide the possibility of initializing the system in an objective way due to its sensitive dependence on initial conditions; see Appendix A.

In such a case, the notion of climate must remain subjective. Climate predictions can then, at best, be treated similarly to probabilistic (ensemble) weather forecasts (Gneiting and Raftery, 2005), i.e., probabilities can be associated with different outcomes based on the (continually evolving) present state of the dynamical variables and the uncertainty in this knowledge (leading to what is discussed by Stainforth et al., 2007; Daron and Stainforth, 2013; DelSole and Tippett, 2018; de Melo Viríssimo et al., 2024). However, this approach makes the comparison of past and future climates ambiguous, and impedes, e.g., a consistent evaluation of possible future climates as time passes; cf. Appendix A still. An alternative is to stick to fixed conventions and corresponding protocols for initialization.

---

[18]This ensemble is infinitely large in principle but is sampled by a finite number of members in numerical modeling.





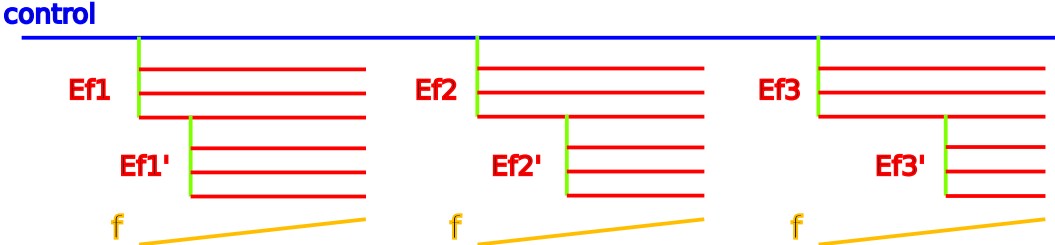

**Figure 3.** Illustration of the initialization scheme proposed in the main text. The horizontal axis stands for time. The vertical axis differentiates between ensemble members which are represented by horizontal lines; these ensemble members are grouped according to the labels. The blue realization (i.e., the control run) is unforced, whereas the red ensemble members are subject to the forcing scenario labelled by f and schematically represented in orange beneath the corresponding ensemble members. Note that the forcing scenario is continued without any interruption (e.g., any kind of restart) when initializing the ensembles with a label with a prime (Ef1', etc.). Vertical light green lines indicate minor perturbations. Unforced ensembles are not covered by this schematic.

We must recall once more that "sufficient" predictability in terms of slower-converging modes must allow for a "complete" convergence in faster-converging modes and during the time span of interest, where "complete" refers to an accuracy required by some practical aspect. As explained in Section 3, a high degree of robustness is expected against which practical aspect is chosen. Even like this, the choice of the practical aspect may be regarded as a subjective element in the definition. However, the resulting climate depends on it only in the sense that it will have an associated *precision* up to which it is defined, in the sense that climates originating from different initializations may scatter within the range of accuracy up to which convergence is required to take place. This is thus an attribute through which the comparability of climates defined in different studies can be controlled or at least assessed[19]; other subjective factors do not give rise to such an attribute.[20]

### 5.2 How to proceed to applications

#### 5.2.1 An initialization scheme

In a given model subjected to a given forcing (i.e., a given form of time dependence), it is actually expected to be possible to decide if a (practically) unique probability density generally appears in a given variable or (derived) observable within a time span of interest, and to decide whether the state of slower-converging modes is relevant in an affirmative case. We propose the following ensemble initialization scheme, pictured in Fig. 3, for this purpose.

In the first step, a single ensemble, Ef1, is initialized by small perturbations of a model state taken from a long control run. The trajectories of Ef1 are then integrated under the forcing of interest [typically the historical forcing followed by some future

---

[19]This is so even if different quantifiers of distance are used for defining the climates to be compared, since any quantifier can be evaluated in a posterior way.

[20]From a philosophical point of view, one could argue that fixing the time span of interest is also subjective. However, we regard it as part of the research question, similarly as the subjective choice of considering the Earth rather than some other planet. Of course, the question of how to define climate can be posed for different time spans or different planets as well. If we like, these are also attributes of a climate.





scenario such as SSP3-7.0 (Gidden et al., 2019)]. The next step is the initialization of a second ensemble, Ef1', by perturbations of a model state taken from an arbitrary member of the first ensemble, Ef1, similarly as in Herein et al. (2023). The delay in its initialization should be around the shortest time scale for which one may first be interested in convergence and uniqueness.

Then the same procedure is repeated for a few further time instants of the control run, resulting in additional ensemble pairs (Ef2,Ef2'), (Ef3,Ef3'), etc., such that Ef$n$ is separated from Ef$(n-1)$ more than the length of the time span of interest (thereby sampling different states of modes with comparable convergence times at least). However, the pairs with subsequent indices $n$ are different in that the delay in the initialization of Ef$n$' with respect to Ef$n$ increases with $n$ until the full time span of interest is approached (at least).

In terms of a given variable, uniqueness can be assessed by evaluating whether Ef$n$' converges to Ef$n$, for various indices $n$, with regard to a practical aspect. In case Ef$n$' converges to Ef$n$ for any arbitrary $n$ with a shorter time scale (approximate $e$-folding time) than the delay between Ef$n$' and Ef$n$ for the given $n$, a "full" convergence with that time scale ensures uniqueness *if* "full" convergence is also observed up to some index $m$ with Ef$m$' separated from Ef$m$ by that "full" convergence time at least. The largest such $m$ will define the longest time span (perhaps the time span of interest or, if studied, even longer) for

which climate can be defined with the given convergence time.[21] Once convergence is noticed to take considerably longer (for indices beyond that just-mentioned largest $m$), it indicates unpredictability arising from modes with comparable time scales of convergence and precludes uniqueness for correspondingly long times.[22]

By intuition, we suspect that investigating convergence to an ensemble that has already begun spreading out from localized initial conditions is technically easier, especially from the point of view of slower-converging modes, than to one initialized

at the same time just localized around a different point in the phase space; this is one advantage of initializing Ef$n$' from a member of Ef$n$ during an active forcing scenario rather than using a different time instant from the control run to initialize Ef$n$' at the beginning of the scenario as well. This is also beneficial because properties of the convergence during the bulk of a forcing scenario may be different and have more practical relevance than those just after switching the scenario on. In fact, by virtue of the increase of the delay with $n$, different epochs within the scenario, possibly with different convergence properties,

are covered by our scheme.

Having said that, however, we must note that our approach is not fully perfect as it does not explore such properties systematically and, at the same time, uses convergence with those possibly different properties for the assessment of the longest time span of a sound definition. In fact, it would be desirable to initialize many ensembles for each $n$ at increasingly later time instants within the scenario, thereby introducing a new index serving for an assessement similar to but (by using later-

initialized ensembles as additional references for convergence) more general than the one described earlier. Our scheme has been designed in the hope, supported by some numerical experience, that convergence properties are usually not considerably different in different epochs of a scenario.

Scenarios have a finite length. Even if it may reach beyond some given "time span of interest", an assessment of arbitrary slower-converging modes [whether their existence or relevance for a given variable, implying issue (i) in an affirmative case]

---

[21]Actually, this longest time span provides an upper bound on the longest possible time span of interest that is meaningful to choose in the given study.

[22]Beyond the interval of sufficient predictability, the "diverging" realizations of the slower-converging modes define different possible climates.



is only possible with the help of a control run. This is what can be performed by evaluating the convergence of ensembles Ef$n$ with *different* indices $n$ to each other.

All of what is discussed above needs a word of caution: certain ensembles of the many may happen to be initialized from very similar states of some mode by chance, which will make the effect of this mode perhaps even invisible when considering the convergence of these ensembles to each other. Without actually knowing these modes, this risk is unavoidable. It can be 440 mitigated by increasing the number of initialized ensembles and carefully checking any suspicious case.

On the other hand, if convergence of Ef$n$' to Ef$n$ fails for a few but only few values of $n$ without being clustered to high $n$, it presumably indicates the problem of regime transitions [a violation of condition (b)].

It is easy to amend the above scheme to study issue (ii). For this purpose, the ensembles Ef$n$ should be compared with ensembles generated with identical initialization but with no forcing (Eu$n$; 'f' and 'u' stand for forced and unforced, respec-445 tively). Note, however, that model drift (Gupta et al., 2013) has the same effect from this point of view as a possible predictable evolution of slower-converging modes; but, at least, forced changes can always be discerned.

Our proposed initialization scheme is different from that of Hawkins et al. (2016), Hogan and Sriver (2019), the CanESM2 large ensemble (Kirchmeier-Young et al., 2017; Singh et al., 2022) and the CESM2 large ensemble (Rodgers et al., 2021) in that *pairs* of ensembles are initialized by perturbations in association with each model state taken from the control run, thereby 450 enabling assessment of convergence to a (possibly conditionally) unique probability density. Timing of initialization is also important; different model states for different pairs are taken from far away time instants of the control run so that slower-converging modes, with respect to the time span of interest, can also be studied. Most of the cited studies target a century or so but apply a separation of just 50 years. Note that model drift (as in Singh et al., 2022) is not at all ideal for assessing the impact of slower-converging modes on faster-converging ones due to its inherently artificial nature.

**5.2.2 Final remarks**

We emphasize here once more that the actual conclusions drawn from an investigation like the one described right above will generally depend on the particular choice of variable to study. This is a further aspect, besides the accuracy of convergence, that is determined by the particular goal of a study. This does not preclude an objective definition either; it just means that a well-defined "climate of different variables" generally applies under different conditions.

As discussed earlier, what we call here a predictable context (the state of slower-converging modes) is also an objective factor setting the scene for climate. Preparing climate projections would require its incorporation into initialization according to the present state; in fact, such an initialization would be useful regardless whether climate can be defined uniquely or not. However, if appropriate observations are not available (as is presumably the case for the deep ocean), generating ensembles from various such states is important for the purpose of mapping out different possibilities permitted by the system, each 465 defining a different climate.[23]

---

[23]The picture could presumably be made much simpler in studies concerning time scales much longer than a century (e.g., paleoclimatic ones), in which convergence in terms of all of the relevant modes may be possible to ensure, so that a single climate can naturally be defined. At the same time, it may also





One undesirable property of any ensemble-based definition of climate is its inaccessibility in single realizations (including the observed evolution of the real Earth system), but, strictly speaking, such an accessibility is not conceptually required in a probabilistic framework [unlike Werndl (2016) suggests].

By such a definition, an important deviation from standard terminology becomes necessary: as long as the underlying prob-
ability density remains (practically) unique, internal variability characterized by time scales shorter than the time scales of slow convergence will not be a source of uncertainty in the description of climate; not even when its future projections are considered. Instead, the very definition of climate should just be a full description of the (possibly conditional) statistics of the system in terms of the mentioned probability density, including the statistics of the kind of internal variability just mentioned.

In practice, this means that all statistical quantifiers should be evaluated with respect to the ensemble. This also implies that
there is no need to "invent" newer and newer ensemble-based statistical quantifiers [such as correlation coefficients (Herein et al., 2017), empirical orthogonal functions (Haszpra et al., 2020), etc.] individually, one by one: instead, one should just follow this "recipe" for the evaluation of *any* quantifier. This does not exclude evaluating ensemble-wise statistics of statistical quantifiers evaluated over time intervals in single realizations ("interval-wise taken" ensemble statistics in Drótos et al., 2015) — however, evaluating a statistical quantifier over a time interval and taking its ensemble mean is not the correct way to learn
about the given statistical quantifier from a probabilistic point of view. This is related to the violation of Birkhoff's ergodic theorem in a system with explicit dependence on time (Drótos et al., 2016), the manifestation of which was already recognized by Daron and Stainforth (2013); as a practical example, sources of nonergodicity for teleconnections as cross-correlations are analyzed in Bódai et al. (2022).

*Code and data availability.* The data presented in Figs. 1 and 2 are from Drotos (2022). The data presented in Fig. C1 were created by Bodai
485 (2022).

## Appendix A: Deficiencies of a definition relying on conditioning on observations

Conditioning the definition of climate on observations of the past may not generally be satisfactory, since it involves two kinds of ambiguity. First, such a climate depends (practically continuously) on how far in the past the observations (more precisely, initial conditions compatible with the observations) are prescribed: if the lead time may be chosen on a case-by-case
basis, we will end up with "personal climates" as discussed in Section 1, and there will be no qualitative difference between the notion of climate and that of probabilistic weather forecast. Second, such a climate will depend on the precision of observations: improving precision may narrow down the magnitude of its internal variability. Uniqueness, from the point of view of objective factors, will be lost due to these dependences.

---

be desirable to conform with definitions created with the purpose of characterizing climate and its change within a century. In fact, the time evolution of slow-converging modes can be included as a forcing if corresponding proxies are available.





These undesired dependences will disappear in practice if there is a considerable separation of time scales of convergence
between the processes (modes) intended to be included in the internal variability of climate and those intended to be excluded
from it, *and* if initialization is chosen sufficiently far (relative to the fast-converging former processes) but not too far (relative
to the slow-converging latter processes) in the past. However, we will then recover the conditional definition introduced in
Section 3, and initialization by observations (assuming it is possible) will be relevant only for the slow-converging processes
(providing the predictable context). The conditional definition motivated by the concept of snapshot attractors and that of
Stainforth et al. (2007); Daron and Stainforth (2013); DelSole and Tippett (2018); de Melo Viríssimo et al. (2024) based on
observations become special cases of each other in this situation. Note, however, the ambiguity that may arise about how to
choose the predictable context if no reliable instrumental records are available. See also Section 5.2.2 in this respect.

## Appendix B: Identifying time scales of convergence

We discuss here how the spectral theory of transfer operators (Lasota and Mackey, 1994; Slegers, 2019) should enable one to
identify time scales of convergence.

Let us first consider an autonomous system, i.e., one without any explicit dependence on time, hypothetically describing a
stationary climate:

$$\dot{\mathbf{x}} = \mathbf{F}(\mathbf{x}), \tag{B1}$$

where $\mathbf{x} \in X$ represents the vector composed of all dynamical variables of the system, with $X$ being the $d$-dimensional phase
space spanned by these vectors, and $\mathbf{F}$ defines the dynamics, which is assumed to be dissipative. Let $\mathcal{P}_{t_0}^t$ denote the Ruelle–
Perron–Frobenius or transfer operator associated with the dynamics between time $t_0$ and $t_0 + t$ with $t \geq 0$ and defined with
respect to the Lebesgue measure of $X$: the time evolution of probability densities $f$ defined on $X$ is described by the action
of operators $\mathcal{P}_{t_0}^t$ on $f$ with different values of $t \geq 0$. If the operators $\mathcal{P}_{t_0}^t$ are quasi-compact (i.e., if they have a finite number
of isolated eigenvalues outside the essential radius, which we hope to be the typical case), the time evolution of an arbitrarily
initialized probability density $f$ from the class of square-integrable functions on $X$ can be decomposed as

$$\mathcal{P}_{t_0}^t(f) = \sum_{i=1}^{N} c_i \lambda_i^{t/T} \varphi_i + r(t), \tag{B2}$$

where $c_i$, $i \in \{1, \ldots, N\}$ are constant coefficients depending on $f$ (such that $c_1 = 1$), $T$ is the unit of time, $N$ is the number of
isolated eigenvalues of the operator $\mathcal{P}_{t_0}^T$, $\lambda_i$ is the $i$th of these eigenvalues with $\lambda_1 = 1$ and $|\lambda_i| < 1$ for all $1 < i \leq N$ (note that
these eigenvalues are generally complex and come in complex conjugate pairs), $\varphi_i$ is the $i$th eigenfunction of the same operator
(note that $\varphi_1$ is the density of the natural probability measure), and $r(t)$ is a residual decaying faster than $\lambda_N^{t/T}$ (Györgyi and
Szépfalusy, 1988; Chekroun et al., 2014; Slegers, 2019; Navarra et al., 2021). If there are multiple attractors, which have
separate basins of attraction in the phase space, the decomposition (B2) applies separately to densities $f$ with an initial support
falling in a single basin (Tantet et al., 2018a).



Eq. (B2) means that $f$ converges to the density $\varphi_1$ of the corresponding natural probability measure, and the convergence

proceeds according to exponential terms with different time scales of decay, as per the real parts of the eigenvalues and accompanied by oscillations as determined by the imaginary parts, and a fast-decaying residual. The exponential contributions with different decay time scales may then form the basis of the conditional definition of climate: if the separation between decay time scales is sufficiently large, "complete" convergence is possible on one time scale without an influence from processes with longer convergence time scales.

A time scale of convergence may (possibly) be associated with a particular system component according to properties of the corresponding eigenfunction. Furthermore, the multivariate density shows how much fluctuations in different system components are correlated.

An important remark to be made here is that time evolution associated with a given eigenfunction may very well happen to be fast according to the oscillatory part, even if the corresponding decay is slow. In such a case, fast time evolution is *predictable*,

as the autocorrelation function can be decomposed in terms of the same modes (one may refer, e.g., to Corollary 1 in Chekroun et al., 2020, which addresses a similar setup); cf. Section 3.

While the above considerations concern autonomous systems, generalization to periodically forced systems is easy by identifying the unit $T$ of time with the period of the forcing. In this case, Eq. (B2) will hold for $t = nT$ with $n \in \mathbb{N}$. Similar results also exist in systems with nonperiodic dependence on time (Froyland et al., 2010), but the eigenvalues are not constant in this

case. Although tippings (Ashwin et al., 2012), especially crises of corresponding stationary chaotic attractors (Tantet et al., 2018a, b), may lead to complications at least for some eigenvalues, if most of the eigenvalues vary moderately enough, as might be expected under forcing scenarios relevant to century-long studies, these eigenvalues remain informative about the time scales in the system (cf. Tantet et al., 2020). An attempt for computing an approximation of the spectrum of the relevant transfer operator is nevertheless beyond the scope of this article.

We mention that more sophisticated and, at the same time, more easily implementable techniques also exist for identifying some kinds of time scale separation; see, e.g., Froyland et al. (2014, 2016), and cf. Lucarini and Chekroun (2023). In contrast, our application appears to necessitate a very specific notion, for which these techniques are not sufficient: an explicit gap in the real part of the eigenspectrum.

**Appendix C: Regime transitions and uniqueness in slow-fast systems**

For simplicity, we use a slow-fast system meant according to traditional terminology (Kuehn, 2015). In particular, we use the stochastic (in the Itô sense) equations of motion

$$dx = (-x^3 + 2x)dt + \sigma_{xx}dW_{xx,t} + \frac{1}{\epsilon}\sigma_{xy}y dt, \tag{C1}$$

$$dy = \frac{1}{\epsilon^2}(-y + cx)dt + \frac{1}{\epsilon}\sigma_{yy}dW_{yy,t}, \tag{C2}$$

to demonstrate that an ensemble initialized by small perturbations during a regime transition cannot represent climate in the

absence of its uniqueness on the short term. A time scale separation between the slow $x$ and fast $y$ variables is achieved



as $\epsilon \to 0$ (Wouters and Gottwald, 2019). To represent a practical, realistic situation, we set $\epsilon = 0.2$. The slow subsystem is characterized by a symmetric quartic polynomial like in Bódai (2020). It is perturbed both by some noise ($\sigma_{xx} > 0$) and the fast subsystem ($\sigma_{xy} > 0$), which latter features (internal) variability generated by white noise ($\sigma_{yy} > 0$) (Hasselmann, 1976; Wouters and Gottwald, 2019). Effectively, the time scale separation between $y$ and some even faster $z$ whose governing

equation is eliminated is readily represented in the stochastic model. The noise $\mathrm{d}W_{xx,t}$ is the slow subsystem's own internal variability, unaffected by the fast subsystem. The fast variable is affected by the slow variable ($c \neq 0$), which is why when climate cannot be defined with respect to the slow variable, it carries over to the fast one. We use the parameter values of $\sigma_{xx} = 0.6, \sigma_{xy} = 0.4, \sigma_{yy} = 1, c = 0.2$. The equations are integrated by the Euler–Maruyama integration scheme, using a time step of $\Delta t = 0.02$.

The quartic polynomial represents a double-well potential function and so gives rise to a saddle-type unstable fixed point, i.e., a saddle, at $x = y = 0$ in the unperturbed ($\sigma_{xx} = \sigma_{yy} = 0$) 2D system (C1)-(C2), also called a Melancholia state in Lucarini and Bódai (2020), whose stable manifold makes a finite angle with the $x = 0$ line owing to the coupling $\sigma_{xy} > 0$. With a weak perturbation (in the sense of Bódai, 2020), infrequent transitions between the potential wells, across the Melancholia state, take place in terms of a long single-realization "control" run.

In order to examine the uniqueness of the converged ensemble, we initialize a pair of ensembles at some state of the control run. Namely, we "perturb" the fast variable as $y + \delta y$, $\delta y = +1$ for one ensemble and $\delta y = -1$ for the other ensemble.[24] We do so in order to have two markedly different[25] ensembles initially — the difference of the emanating densities has to disappear on the fast time scale to allow for a unique definition of climate. When the dynamics is deterministic, a minute mismatch of the initial conditions within each of the ensembles — at least with respect to $z$ — is needed for the spread of that ensemble.

In our stochastic modeling, however, it is clearly not required. The various different realizations of the ensemble members are generated by various different realizations of the Wiener process $W_{yy,t}$.

Given the said inclination of the stable manifold, which is the basin boundary of the unperturbed system, the opposite perturbations $y \pm \delta y$, even if taken in the $y$-direction only, could already achieve the placement of the two initial conditions in the different basins of attraction, provided that the current $(x, y)$ was not far off from the basin boundary. Then, ensuing

perturbed ($\sigma_{yy} > 0$) realizations will much more likely end up in the near future in the potential well/regime where they started out from. As only the fast variable is perturbed to initialize an ensemble, the same single realization of $W_{xx,t}$ of the control run is used for all of the ensemble members. In effect, this gives rise to nonautonomous dynamics, i.e., explicit time dependence in the system. Therefore, it is the stable manifold of the corresponding snapshot saddle that will actually control the transitions (Bódai et al., 2013).

When we initialize the two ensembles during a transition (see the middle row of Fig. C1), the ensemble means of neither the fast nor the slow variable, $\langle y \rangle$ and $\langle x \rangle$, converge but remain well separated, indicating the lack of uniqueness, at least on the time scale of the fast variable $y$. I.e., the realized ensemble will depend on the particular initialization of the fast subsystem.

---

[24]Such a "perturbation" of a state for the purpose of *generating* initial conditions for trajectories is not to be confused with the "dynamic perturbation" of trajectories under the evolution equations as a result of $\sigma_{xy} > 0$, etc., in (C1)-(C2).

[25]Note that the standard deviation of $y$ is comparable to $|\delta y| = 1$, and so the applied perturbations can correspond to extreme opposite, or, rather different, states of the fast process.





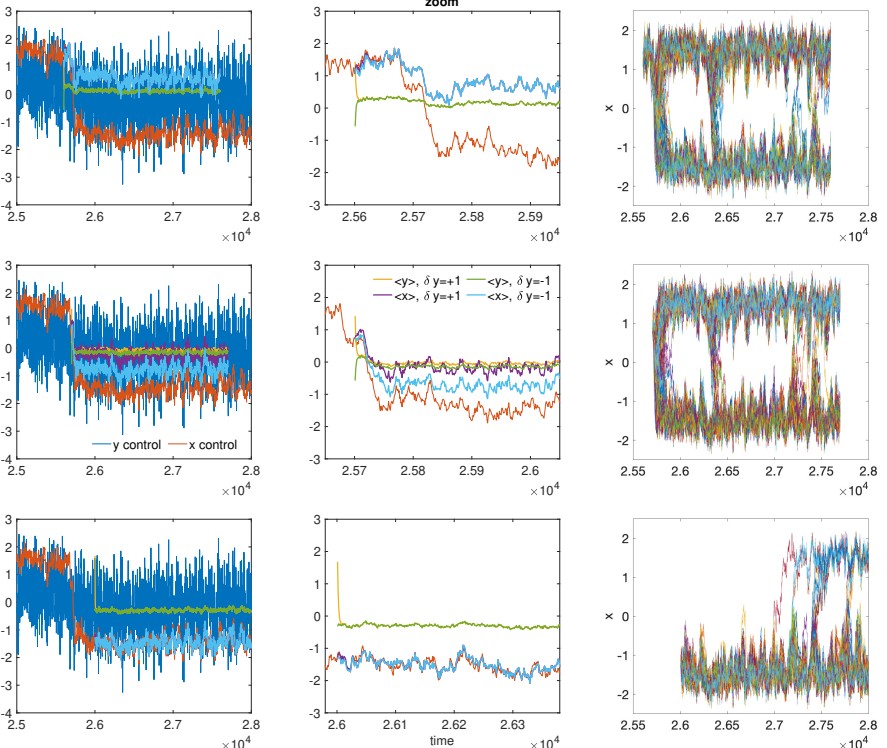

**Figure C1.** Initialization during and away from a regime transition precludes and supports the definition of climate, respectively. Two ensembles are initialized by perturbing the fast variable as $y + \delta y, \delta y = +1$ or $-1$ (see main text), upon which the ensemble means $\langle y \rangle$ or $\langle x \rangle$ are plotted in diagrams on the left to examine convergence, using 1000 ensemble members in each ensemble. Diagrams in the middle column provide zoomed pictures of those on the left with respect to time, excluding the $y$ control time series, for the better visibility of uniqueness. The legend annotations in two panels apply to all in the left and middle columns. On the right, spaghetti diagrams of all of the 1000 slow time series are given for one of the ensembles ($\delta y = +1$). Data were produced by code by Bodai (2022).

This indeed prevents any of these to objectively represent climate. On the contrary, when we initialize the two ensembles sufficiently far away from a transition, whether before or right after it (see the top and bottom rows of Fig. C1, respectively),
the ensemble means converge on the time scale of the fast variable, indicating uniqueness. Due to the "forcing" of the slow subsystem, $W_{xx,t}$, clusters of trajectories (a subset of all of the trajectories because of $\sigma_{xy} > 0$) suffer a regime transition in a coordinated manner, which is imprinted on the evolution of the ensemble means. That is, had the toy model reflected realistic characteristics of the climate system, regime transitions of the slow subsystem could give rise to the delineation of the concepts of climate change and forced response.
The case of $\sigma_{xx} = 0$, when the slow subsystem does not have an internal variability in isolation from the $y$ system component, is qualitatively similar: there is no uniqueness on the fast time scale in association with initialization during a regime transition of $x$.



*Author contributions.* Conceptualisation: GD; writing—original draft preparation: mainly GD; writing—review & editing: mainly TB; every other task: shared in various proportions.

*Competing interests.* The authors declare that they have no conflict of interest.

*Acknowledgements.* T. Tél deserves special thanks for his role in earlier work forming the basis of the present article. The contribution of M. Herein to preparing and running the numerical simulations is gratefully acknowledged. GD is indebted to G. Froyland for his invaluable help by providing detailed explanations with regard to the spectral theory of transfer operators. Important insights into this theory are also due to the kind help of M. Chekroun and G. Gottwald. Zs. Mihálka interpreted some of the above information to GD, which is highly

appreciated. Useful discussions with C. Franzke, E. Hernández-García, A. Navarra, K. Rehfeld, B. Sándor, D. Stainforth, A. Tantet and T. Tél are acknowledged as well, along with helpful comments on the manuscript by T. Tél and M. Vincze. GD acknowledges financial support from the National Research, Development and Innovation Office (NKFIH, Hungary) under grant K125171, as well as from the European Union (European Social Fund and European Social Fund Plus) and the Government of the Balearic Islands through the Margalida Comas and Vicenç Mut postdoctoral fellowships (grant numbers PD/020/2018 and PD-035-2023, respectively). TB was supported by the Institute

for Basic Science (IBS) under grant IBS-R028-Y1 and by the Research Excellence Programme of the Hungarian University of Agriculture and Life Sciences.



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
