# Peer review of "On defining climate by means of an ensemble"

_EGUsphere, 2025_

## Referee Comment (RC1)

*A Review of*

**On defining climate by means of an ensemble**

*by Gábor Drótos and Tamás Bódai*

**General.** This paper is a wide-ranging discussion of various approaches to describing, simulating and possibly defining climate. It is rather descriptive, while attempting to be normative. I found it hard to read and will try to offer some suggestions on how to make it more readable as opposed to being all-inclusive.

Overall, the paper does contribute to bringing to the attention of the *ESD* readership — and to a fairly rapidly increasing community of mathematicians and physicists interested in climate and its change — concepts and methods from the theory of nonautonomous dynamical systems (NDSs). This theory is clearly well-adapted to the description, understanding and prediction of the way that time-dependent forcing or coefficients affect a system that has both chaotic and random elements, like the climate system. A major problem in applying this fairly novel mathematical theory effectively to the difficult problems at hand is the use of different language by two distinct communities of practitioners: some that rely on a somewhat fuzzy "physical" background and those that try to use efficiently the advances of the rigorous mathematical theory. Clearly the authors are respected members of the former community and the reviewer belongs to the latter.

The recommendation is publication after improvements.

**Major comments**

*Philosophical issues.* The reviewer (hereafter MG, to avoid the pretentious "I".) has as much of a philosophical bend as the authors. As such, he would like to recall the difficulties that Ludwig Wittgenstein already pointed out in the communication among different "language communities," into which he definitely included scientific communities. This is certainly the case in the communication among the physical scientists of IPCC's Working Group (WG) I and the socioeconomic experts of WG II and WG III. But it is also the case in the problems at hand, among the authors, who only quote as their dynamical systems bibles Tél & Gruiz (Cambridge UP, 2006), a fine undergraduate book, and Ott (Cambridge UP, 1993), published before the evolution of NDS theory.

The refusal to take to heart and to bed recent books like Caraballo & Han (Springer, 2016) or Kloeden & Yang (World Scientific, 2020) — which not only have the word "nonautonomous" in the title but also treat the latest developments in NDS theory — is not helpful in overcoming the Wittgenstein-type miscommunication between the two communities. It might also be worth recalling Galilei's words on the role of mathematics in understanding the world and how it functions: "Philosophy is written in this grand book — I mean the Universe — which stands continually open to our gaze, but it cannot be understood unless one first learns to comprehend the language and interpret the characters in which it is written. It is written in the language of mathematics, and its characters are triangles, circles and other geometrical figures, without which it is humanly impossible to understand a single word of it." (translated from "Il Saggiatore, vol. VI, p. 232" in Galileo, *Opere*, A. Favaro, Ed., Barbèra, Firenze, 1929–1939). One does not have to spend one's life proving theorems, but it is helpful to have an idea of what they are about.

MG appreciates the reference in the paper to the work of Charlotte Werndl, but does not find that the mixture of trying to proceed in a similar mode in this paper, on the one hand, with ignoring entirely the major ingredient of rigorous mathematics, on the other, is helpful in advancing the subject.

*Practical issues.* It is not clear to MG in which way what is purported to be a definition of climate is really different from the ensemble simulations that are practiced by the IPCC's Assessment Reports and the successive Coupled Model Intercomparison Projects (CMIPs) that they rely upon. It is nice to do to small models what has become the *modus operandi* in dealing with very large ones. But this is just a mode of describing climate and hoping to predict its change, rather than a definition thereof.

**Recommendation**. Shorten and sharpen the text of the paper and change the title to "**Describing climate by means of an ensemble.**"

Michael Ghil

P.S. In trying to properly place the problem of time-dependent forcing of a chaotic and random system like the climate system among those posed by the system's nonlinearity in general, it might be worthwhile citing

Ghil, M., 2019: A century of nonlinearity in the geosciences, *Earth & Space Science*, **6**, 1007–1042, doi: 10.1029/2019EA000599,

and its several "lamp posts."

Likewise, in another EGU journal than the one to which this paper is submitted, namely *NPG*,

Ghil, M., 2020: Review article: Hilbert problems for the climate sciences in the 21st century – 20 years later, *Nonlin*. *Processes Geophys*., 27, 429–451, https://doi.org/10.5194/npg-27-429-2020.

---

## Referee Comment (RC2)

**Review of "On Defining climate by means of an ensemble" by G. Drótos and T. Bódai.**

The question of defining the probabilistic properties of the climate system under changes is addressed in this manuscript. The idea is to relax the constraints of introducing initial conditions in the far past to construct pullback attractors, the reason being that it necessitates very long model integrations, in particular in multiscale systems, in order to reach the proper pullback attractor. After introducing the key concepts of attractors under external forcing, the authors make an attempt of building a pragmatic scheme to check if the main statistical properties of the reached distribution are indeed robust when starting from another set of states at another moment of the system's evolution. The question is important and is worth to be published in Earth System Dynamics. There are however two points that, I think, should be further elaborated. These points follow:

The authors use a model, PlaSim, which is an atmospheric model coupled to a slab ocean. This choice is very restrictive as we know the importance of interactions between the atmosphere and the other components of the climate system (ocean, cryosphere, land/vegetation…). This implies that a low-frequency variability can emerge in reality which is not accounted for by the PlaSim model. An example of such an emergence of low-frequency variability in the climate system due to the interaction between several systems having different time scale characteristics is provided in Vannitsem et al (2021) where a reduced-order coupled ocean-atmosphere extratropical model is forced by a simple low-order recharge-discharge oscillator describing the evolution of the Tropical Pacific. The introduction of the Tropical forcing induces an emergence of non-trivial low-frequency variability in the extratropical model which are then present in the Pullback attractor. This feature suggested us that the convergence toward the Pullback attractor should at least be associated with the slowest time scales of the system under investigation. As mentioned by the authors, there is a possibility to build some "conditional" attractor provided there is an enough separation of scales, but when new time scales emerge related to the interactions (or resonances) between multiscale processes, the question becomes much harder. This question should be discussed in the present work (probably in Section 4) to have a full overview of what are the caveats and difficulties that may be encountered in defining the statistical properties of the system under climate change.

Related also to the separation of time scales and the definition of climate, there is a very interesting separation that was introduced in Lovejoy (2015). In this paper, Lovejoy defined regimes at different time scales, related to their scaling properties. In this context, the question of convergence of the statistical properties (mean, variance…) at the different time scales is discussed. There is one particular time scale range (10 days to 50 years), the macro-weather for which the convergence of the mean seems possible while for other periods, it seems to diverge. This could also help you in figuring out what are the time scales for which your definition of attractors may apply. A discussion around this question should be useful.

I also agree with the change of title and recommendations of Reviewer 1.

**Some minor points**

Line 12: "In" should be "If" I guess.

Line 30. The authors should provide examples on the different definitions of "climate" if any.

Line 36. Missing reference at the end of the line.

**References:**

Lovejoy, S. A voyage through scales, a missing quadrillion and why the climate is *not* what you expect. *Clim Dyn* **44**, 3187–3210 (2015). https://doi.org/10.1007/s00382-014-2324-0

Vannitsem, S., Demaeyer, J., & Ghil, M. (2021). Extratropical low-frequency variability with ENSO forcing: A reduced-order coupled model study. *Journal of Advances in Modeling Earth Systems*, 13, e2021MS002530. **https://doi.org/10.1029/2021MS002530**

---

## Referee Comment (RC3)

**Referee report for "On defining climate by means of an ensemble" by G. Drótos and T. Bódai**

September, 2025

**1  Summary**

The main aim of this paper is to provide a definition of climate, as highlighted by the title and the abstract. Accordingly, this will be the central concept on which this feedback will focus.

In Section 2, the authors begin by considering an intermediate-complexity climate model as an illustrative example. Then, climate is introduced as the probability distribution obtained from the time evolution of ensemble trajectories. The idea is that this ensemble approximates what the authors call the "natural probability density," which is the abstract target of their definition.

In Section 3, the authors point out that two aspects must be taken into account: (i) the time evolution of the ensemble should be long enough to allow convergence to occur, and (ii) the previous point implies that, when the study is performed on a time interval much shorter than the one needed for convergence, one must take care of the slow variables. If the separation between the slow and fast time scales is large, then the definition of climate still holds, provided that it is considered as a conditional definition, i.e. depending on the fixed values assumed by the slow variables.

Furthermore, the authors identify two main problems that can undermine the definition of climate if either of the following conditions is not satisfied: (a) there is a large separation between the slow and fast time scales, and (b) the slow variables are initialized far away from a regime transition.

In Section 4, the authors review how the limitations to the definition of climate presented in Section 3 appear in real examples.

Finally, in Section 5.1, the definition of climate is recalled, and an initialization scheme is proposed to decide whether the uniqueness of the probability density holds and whether the slow variables affect the targeted probability density.

**2 Evaluation**

In my view, the paper does not introduce a sufficiently new or significant contribution to the community. For example, consider the closing definition of climate proposed by the authors (lines 358–372): "an operational definition of climate [...] might rely on a decadal-scale convergence of an ensemble [...] to a (practically) unique but time-dependent probability density: this density could be identified with climate." My question is: what is the novelty here? While I appreciate that the authors emphasize the difficulties that can arise if conditions (a) or (b) are not satisfied, they do not seem to offer a way of overcoming these issues; instead, they incorporate (a) and (b) as assumptions in their definition of climate.

To phrase my concern in a different framework: assume that the intermediate-complexity model presented by the authors is given by a non-autonomous stochastic differential equation

$$dX_t = b(X_t, t)\, dt + d\eta_t,$$

where $(X_t)_t$ is a stochastic process in $\mathbb{R}^d$, and $(\eta_t)_t$ models the noise. The authors appear to be saying—if I have understood correctly—that if the time-varying family of invariant measures $(\mu_t)_t{}^1$ of the SDE is unique, then that is "the climate." My point is that this is already a well-established concept in the literature. I would therefore encourage the authors to state more explicitly what the novel contribution of their paper is.

In addition, echoing a point already raised by another referee, I would suggest that the manuscript be revised for greater precision in the formulation of its arguments and statements. I acknowledge that this comment may be influenced by my own mathematical background, but I believe that improving the clarity of exposition would benefit a broad readership.

**3 Typos**

The paper is essentially free of typographical errors. I list below the few that I noticed:

- Line 12: replace "in" with "if".

- Line 32: should read "characteristic of".

- Lines 499–501: the sentence "The conditional definition motivated [...] in this situation" has a singular subject ("conditional definition") but a plural verb ("become").
* * *
[1] I refer, for instance, to [DSF24, Section II.E] for the rigorous definition of the family $(\mu_t)_t$.

**References**

[DSF24] G. Del Sarto and F. Flandoli. A non-autonomous framework for climate change and extreme weather events increase in a stochastic energy balance model. *Chaos: An Interdisciplinary Journal of Nonlinear Science*, 34(9):093122, 09 2024.